# Guillain-Barré Syndrome with Respiratory Failure following Spine Surgery for Incomplete Cervical Cord Injury: A Case Report and Literature Review

**DOI:** 10.3390/medicina58081063

**Published:** 2022-08-06

**Authors:** Wei-Cheng Tu, Shin-Tsu Chang, Chun-Han Huang, Yuan-Yang Cheng, Chun-Sheng Hsu

**Affiliations:** 1Department of Physical Medicine and Rehabilitation, Taichung Veterans General Hospital, Taichung 407, Taiwan; 2Department of Physical Medicine and Rehabilitation, Kaohsiung Veterans General Hospital, Kaohsiung 813, Taiwan; 3Department of Physical Medicine and Rehabilitation, Tri-Service General Hospital, School of Medicine, National Defense Medical Center, Taipei 114, Taiwan; 4Graduate Institute of Sports and Health Management, National Chung Hsing University, Taichung 402, Taiwan; 5Department of Post-Baccalaureate Medicine, College of Medicine, National Chung Hsing University, Taichung 402, Taiwan

**Keywords:** Guillain-Barré syndrome, spinal cord injury, spinal surgery, case report, respiratory failure

## Abstract

Guillain-Barré syndrome (GBS) often develops after a respiratory or gastrointestinal infection. A few cases have been reported on GBS following elective spinal surgery not preceded by an infectious disease. In patients with underlying upper motor neuron disease such as a spinal cord injury, concurrent development of lower motor neuron diseases, such as GBS, could be overlooked. Here, we present an uncommon case of an 87-year-old man with GBS that had developed after an operation for a traumatic cervical spinal cord injury. After surgery, he showed weakness over all four limbs with paresthesia, but he was able to hold a standing position with minimal assistance. Unfortunately, his muscle strength over his four limbs gradually weakened from two to four weeks later, and he became almost completely paralyzed. Cerebrospinal fluid (CSF) studies revealed albuminocytologic dissociation. A nerve conduction study (NCS) indicated an acute axonal polyneuropathy superimposed on chronic sensorimotor polyneuropathy. Thus, the patient was diagnosed with GBS. However, the patient’s family declined immune-modulatory therapy due to personal reasons. The patient progressed into respiratory failure and remained ventilator-dependent before his death three years later. This case highlights the importance of taking GBS into account when postoperative weakness occurs in patients with spinal cord injury, and a worse prognosis if GBS is left untreated.

## 1. Introduction

Guillain-Barré syndrome (GBS), commonly presented with the subtype of acute inflammatory demyelinating polyradiculoneuropathy (AIDP), is the most common cause of acute polyneuropathy in adults [1]. GBS incidence in Western countries ranges from 0.89 to 1.89 cases (median, 1.11) per 100,000 person-years, with a male-to-female ratio of 1.78 [2]. Clinically, GBS often manifests as limb weakness accompanied by diminished deep tendon reflex (DTR) and mild paresthesia. Tetraplegia, dysphagia, and respiratory failure could occur in severe cases. Laboratory studies reveal albuminocytologic dissociation in the cerebral spinal fluid (CSF) is obtained through lumbar puncture in 50–75% of cases, depending on the time after symptom onset [2]. Nerve conduction studies (NCS) and electromyography (EMG) typically indicate that demyelinating polyneuropathy, and axonal type are less common. GBS is usually preceded by infections, such as upper respiratory tract infection, acute gastroenteritis, vaccination, and various viral and bacterial infections [2]. There are some rare case reports on GBS that occurred after elective spinal surgery [1,3,4,5]. Here, we present a case of GBS following surgical intervention for a traumatic cervical spinal cord injury.

## 2. Case Presentation

The patient, an 87-year-old man, presented at our rehabilitation ward with postoperative weakness over both lower limbs. He had a traffic accident, leading to an incomplete type of cervical spinal cord injury. On the American Spinal Injury Association (ASIA) impairment scale, the injury was categorized as grade D. Initially, he had neck pain, bilateral upper limb numbness over bilateral C5-C7 dermatome, and weakness over all four limbs, with a muscle strength grade of 3/5. Preoperative computed tomography (CT) and magnetic resonance imaging (MRI) revealed a fracture in posterior element of C5-C6, and anterior subluxation of C6-C7 with disc posterior herniation compressing the spinal cord (Figure 1). We performed emergent decompression of C6 and C7, and anterior cervical discectomy and fusion (ACDF) of C6-C7 with cervical spine locking plates (CSLP) for fixation. After the operation, he had partial relief of neck pain, numbness, and limb weakness. Rehabilitation programs were then arranged. One week later, his muscle strength over four limbs gradually improved to grade 4/5, and he was able to tolerate a standing position with fair balance. As for his medication history, besides NSAIDs for pain management after surgery, he was also taking metformin, basal insulin for type 2 diabetes mellitus, an alpha blocker for benign prostate hypertrophy, and a proton pump inhibitor for a gastric ulcer. He was a social drinker and had family history of hypertension and diabetes mellitus.

However, two weeks after surgery, his muscle strength of bilateral lower limbs suddenly diminished to grade 3/5 and continued to worsen. A neurologist was consulted to explore the causes of muscle weakness and a series of examinations were made to explore possible conditions, such as electrolyte imbalance, surgical issue of cervical spine, lumbar spine problem, or intracranial lesion. He had no medical history of a gastrointestinal tract infection or respiratory infection before or after the cervical surgery. While laboratory tests (Appendix A) and microbiology studies showed unremarkable findings, ongoing infection was also excluded. Image studies (Figure 2 and Figure 3) indicated no instrumental backout, no new onset of spine stenosis, or intracerebral hemorrhage. A lumbar puncture was then performed under the suspicion of GBS. CSF studies revealed albuminocytologic dissociation, including increased protein levels (167 mg/dL) and an absence of WBC. Blood tests showed normal electrolyte balance (Appendix A).

NCS and EMG examination were also performed. Motor NCS (Table 1) revealed decreased amplitude over bilateral median, tibial, peroneal, and left ulnar compound action motor potentials (CMAPs); decreased conduction velocities over left ulnar and bilateral peroneal CMAPs. An unresponsive F wave was noted over bilateral median and peroneal nerves. Sensory NCS (Table 1) revealed decreased amplitude over bilateral median, radial, and sural sensory nerve action potentials (SNAPs). Needle EMG (Table 2) showed generalized decreased recruitment over bilateral first dorsal interosseus (FDI), left abductor pollicis brevis (APB), left medial gastrocnemius, left flexor carpi radialis (FCR), and right biceps, and no volitional activity over bilateral tibialis anterior (TA) and right medial gastrocnemius muscles. Few polyphasic waves were noted over bilateral FDI, left APB, and left medial gastrocnemius muscles. The above findings suggested an acute axonal polyneuropathy superimposed on chronic sensorimotor polyneuropathy. The diagnosis of acute motor and sensory axonal neuropathy (AMSAN), a rare axonal variant of Guillain-Barré syndrome, was made in the third week after the operation. Immuno-modulatory treatment, such as intravenous immune globulin and plasma exchange, was initially suggested. However, the patient’s family declined the treatment due to concerns of a possible adverse reaction, his advanced age, and for religious reasons. Therefore, only supportive care and bedside rehabilitation programs were provided.

Despite intensive rehabilitation programs, the patient’s motor function continued to deteriorate. Three weeks after the operation, total paralysis with absent DTR was noted below L2 myotome. He also complained about numbness in both feet. Apart from diminished strength of both of his lower limbs, his bilateral upper limbs also showed increasing weakness, a condition that is consistent with the pattern of progressive ascending paralysis. One day later, total paralysis was noted over both arms (Figure 4). Respiratory failure occurred within two weeks after the GBS diagnosis. He was intubated with ventilator support and promptly transferred to the intensive care unit. Later, a tracheostomy was performed after unsuccessful extubation, and he was transferred to the respiratory care ward. He remained in a bedridden status for the next three years and suffered from recurrent pressure sores and pneumonia until his death at our hospice care ward.

## 3. Discussion

Guillain–Barré syndrome is often considered a post-infectious disease as >2/3 of cases are preceded by a respiratory or gastrointestinal tract infection. Pathogens that are commonly identified include Campylobacter jejuni and cytomegalovirus, while Epstein–Barr virus includes varicella–zoster virus and Mycoplasma pneumonia [2,6,7]. Immune responses triggered by infections presumably damage peripheral nerves leading to the development of GBS.

A key feature of GBS is an elevated CSF protein level without pleocytosis, i.e., albuminocytologic dissociation. Compressive myelopathy should be considered, because it may also result in hyperproteinorachia, which reflects damage to axons and neurons [8]. Other causes of increased protein levels in CSF include meningitis, transverse myelitis, and neuromyelitis optica, and often occur along with pleocytosis. In bacterial meningitis, the CSF would present as cloudy in appearance and contain lower levels of glucose. In rare cases of arteriovenous fistulas and paraneoplastic myelopathy, hyperproteinorachia could be seen without pleocytosis [9], and should be carefully differentiated by clinical history and other examinations including MRI, MRA, and cytology. Chronic inflammatory demyelinating polyradiculoneuropathy (CIDP) is another possible condition, if the duration of the disease lasts longer than 8 weeks [10], combined with NCS, indicating demyelinating polyneuropathy. Clinically, it could be difficult to distinguish between CIDP and GBS with poor recovery. An important feature is that CIDP rarely involves severe bulbar or respiratory weakness, while it is more common for GBS to cause respiratory failure.

According to a French epidemiologic study, GBS is associated with certain surgical procedures (adjusted odds ratio, OR = 1.53), especially bone and gastrointestinal tract surgeries (OR = 2.78 and 2.36, respectively) [11]. However, the underlying mechanism between GBS and surgical operation is unclear. Current hypotheses are the following: firstly, during operation, antigens are released leading to autoimmunization [12]. This hypothesis is supported by inflammatory features present in nerve biopsy of patients developing post-operative polyneuropathy [13]. Secondly, during surgical procedures the endocrine stress system is activated, leading to transient immunosuppression resulting in peripheral nerves being attacked by autoantibodies [11,14]. In addition, spinal surgery causes bleeding and inflammatory responses in the spinal cord and spinal canal, potentially triggering further neuro–immune interactions. Increased production of heat stroke protein (HSP70) as triggered by traumatic events may serve as a molecular link between injury, systemic inflammatory cascade, and the subsequent GBS development [15].

Spinal cord injury generally manifests as upper motor neuron signs, such as increased DTR and muscle tone, after patients have recovered from spinal shock. Cases of GBS are categorized by patients with peripheral polyneuropathy typically showing lower motor neuron signs, such as areflexia [2]. In the case of progressive limb weakness after traumatic spinal surgery, multiple conditions need to be considered. For example, acute cerebral infarction, cerebral hemorrhage, undetected associated injury, electrolyte imbalance, and other surgery-related issues. To make a diagnosis of GBS following spinal surgery is challenging, since its symptoms such as paralysis, paresthesia, and pain may arise from spinal cord injury or nerve root compression. In fact, iatrogenic nerve injury due to procedures of ACDF is rare. Common postoperative complications of ACDF and their incidence are the following: transient dysphagia (9.5%), hematoma requiring surgical intervention (2.4%), symptomatic recurrent laryngeal nerve palsy (3.1%), dural penetration (0.5%), esophageal perforation (0.3%), worsening of preexisting myelopathy (0.2%), instrumentation backout (0.1%), and wound infection (0.1%) [16]. In our present case, the time interval between the onset of symptoms and surgery is two weeks, making the possibility for iatrogenic spinal cord injury and postoperative hematoma less likely. Furthermore, absent DTR with symmetric ascending paralysis and newly onset of paresthesia over lower limbs were noted in our patient. His CSF study showed albuminocytologic dissociation. NCS showing systemic demyelinating polyneuropathy or axonal injury supported the diagnosis and helped differentiate various subtypes of GBS. GBS was diagnosed in our patient based on clinical symptoms, CSF study, and NCS findings, despite the atypical absence of prior history of GI tract or respiratory infections. In the initial two weeks after symptom onset, albuminocytologic dissociation is reported in only 60% of CSF patients [17].

A literature review of 33 cases reported between 1987 and 2017 concluded that GBS could develop after various kinds of operations, such as spinal surgery, coronary artery bypass, valvular repair procedure, pelvic and hip surgeries, gastric bypass procedure, colon cancer operation, stem cell transplantation, and organ transplantation [18]. Males, compared with females, seem to be more prone to suffer from postoperative GBS (among these 33 patients: 24 were male and 9 were female). The ratio of male-to-female is even higher in postoperative GBS (2.67) than overall GBS (1.78) [2]. The age of postoperative GBS has a peak in the 50–59 age group, with the youngest case being a 6-year-old boy and the oldest case being a 73-year-old woman [19]. Spine surgery is the most common procedure triggering postoperative GBS, followed by cardiac surgery [18]. The onset time of postoperative GBS varies widely across cases, ranging from 3 h to 1 year [3,20]. Most cases of postoperative GBS (73%) would develop within two weeks at a rapid speed [19,21,22,23,24]. Progressive weakness (94%), paresthesia (73%) and areflexia (48%) are leading clinical manifestations of postoperative GBS [18]. A few patients with bulbar palsy, and autonomic dysfunction were also documented. Approximately one third (36%) of these patients have suffered from respiratory failure requiring mechanical ventilation for survival throughout the worst period of disease progression. Most of these patients recovered fully. Only five patients (15%) died from underlying disease recurrence or multisystem failure.

Cases of GBS were reported following spinal surgeries of cervical, thoracic, and lumbar levels. We had reviewed existing literature on 16 cases (Table 3). Among nine cases of lumbar surgery, six cases involved elective operations for disc herniation [1,5,14] or degenerative scoliosis (from T10 to L5) [4], one involved an emergent procedure for acute cauda equina syndrome [23], and one involved kyphoplasty for traumatic fracture [19]. Two GBS cases occurred following thoracic spine surgery, one spinal canal decompression with fusion for traumatic fracture and one meningioma resection [21,22]. Two cases of cervical disc herniation receiving ACDF and laminoplasty, one case of congenital occipitocervical malformation with disc herniation, and another case of osteochondroma receiving tumor resection [4]. Finneran et al. [25] reported a case of traumatic spinal cord injury (gunshot wound to T4 segment) with an American Spinal Injury Association impairment scale (AIS) grade A, and preserved upper limb strength. All these case reports have nerve conduction findings in support of the GBS diagnosis. AIDP and AMSAN were both identified in different cases. Regarding symptom onset after surgery, 25% start within 6 h, 25% within 1–3 days, and the remaining 50% within 1–2 weeks.

The ratio of mechanical ventilation use during the nadir of symptoms is higher in GBS following spinal surgery (68.8%), when compared with other postoperative surgeries (36%) [18] or overall GBS cases (25%) [2]. After appropriate treatment and supportive care, most GBS patients’ post-spinal operation succeeded in weaning and extubating, including all three patients with traumatic spinal injury [19,22,25].

Our patient was initially diagnosed with a motor incomplete type of cervical spinal cord injury, with AIS grade D. According to the literature, patients with spinal cord injury, AIS grade D, have a good prognosis of walking function, with an overall recovery rate of 100% [26]. Another retrospective study revealed that 80% of patients with AIS grade D, over the age of 50, restore independent walking function [27]. The underlying comorbidity and complications likely contributed to this difference [28]. Furthermore, among patients with bedridden status caused by GBS, 80% of them could restore their walking ability to some extent, 6 months after the onset of symptoms [2]. The prognosis of GBS cases following spinal surgery is also good. Given appropriate treatment of IVIG or plasmapheresis, most patients recover, with minor sequelae on motor or sensory function. Only one patient who had received IVIG treatment became mechanical-ventilator-dependent 9 months later [4]. Our patient was an 87-year-old man with symptom onset of ascending paralysis two weeks after spinal surgery and was diagnosed with GBS two weeks later. He did not receive immunotherapy due to family objection, given his age and religious concerns. Unfortunately, our patient became ventilator-dependent and never restored his motor function. Given the example of our case, the diagnosis of GBS should always be kept in mind for patients who, presenting with postoperative spinal cord injury, develop new onset of decreasing muscle strength together with lower motor neuron signs. With the early diagnosis of GBS, the immunomodulatory treatment should be administered promptly, within two weeks after symptom onset, to reach a better prognosis.

**Table 3 medicina-58-01063-t003:** Summary of published cases of Guillain-Barré syndrome following spinal surgeries.

Author/Year	Lesion	Age/Sex	Surgical Procedure	Onset *	Therapy	Respiratory Failure	Follow-Up (months)	Prognostic Outcome
Chen et al., 2017 [1]	Lumbar degeneration with lumbar stenosis	57/M	Lumbar fusion at L3-S1	9 days	CS, IVIG	Yes	16	Weakness over left diaphragm
Boghani et al., 2015 [3]	Degeneration of the left L4-5 facet and lateral recess stenosis	58/M	L4-L5 right-sided hemilaminotomy	<3 h	PP, IVIG	Yes	12	Paresthesia over lower trunk and legs
Boghani et al., 2015 [3]	L3-4 disc hernation	40/M	L3-L4 lumbar hemilaminectomy	<3 h	PP, IVIG	No	18	Some residual numbness of legs
Huang et al., 2015 [4]	Congenital occipitocervical malformations and Cervical disc heriation	50/M	Occipitocervical fusion (Occipital-C2) and ACDF (C5-6)	7 days	IVIG	Yes	20	Independent ambulation with a cane
Huang et al., 2015 [4]	Cervical spinal stenosis	53/M	C3-6 laminoplasty	3 days	IVIG	Yes	20	Independent transfer from bed to chair
Huang et al., 2015 [4]	Degenerative lumbar scoliosis	69/M	T10-L5 posterior-approach lumbar interbody fusion	2 days	IVIG	Yes	9	Ventilator-dependent
Huang et al., 2015 [4]	Cervical disc heriation	58/M	C4-7 ACDF	3 days	IVIG	No	5	Minor weakness of the intrinsic muscles of the hands
Dowling et al., 2018 [5]	Failed back surgery syndrome	53/F	Revision lumbar decompression and spinal fusion	10 days	IVIG, AAb	No	2.5	Nearly recover to baseline
Rashid et al., 2017 [14]	Recurrent degenerative lumbar stenosis	62/F	Revision lumbar decompression and spinal fusion	11 days	IVIG	Yes	12	Independent ambulation
Battaglia et al., 2013 [19]	L1 vetebral fracture	73/F	L1 Kyphoplasty	7 days	IVIG	No	4	Left side facial palsy, peripheral type
Cheng et al., 2011 [21]	T1-T3 meningioma	59F	T1-T3 laminectomy with tumor resection	6 h	IVIG	Yes	7	Independent transfer from bed to chair
Son et al., 2011 [22]	T12 vetebral fracture	50/M	T12 Spinal canal decompression and fusion	10 days	IVIG	Yes	2	Minor weakness of the intrinsic muscles of the hands
Torregrossa et al, 2021 [23]	L4-L5 disc herniation with acute CES	76/M	L4-L5 microdiscectomy	Immediately	IVIG	Yes	NA	NA
Miscusi et al., 2012 [24]	Chondroma	55/M	Tumor resection at C6-C7 level with laminoplasty	36 h	IVIG, CS	No	3	Progressive proximal-distal recovery of strength in both legs
Finnera et al., 2020 [25]	GSW to spine at T4 level	23/M	Thoracotomy, right upper lobe wedge resection	16 days	PP	Yes	9	Full upper extremity strength, independent use of wheelchair
Sahai et al., 2017 [29]	Lumbar spinal stenosis	52/M	L4-L5 decompression and spinal fusion	17 days	IVIG	No	6	Independent ambulation

* Onset of symptoms after operation. ACDF, anterior cervical discectomy and fusion; CES, cauda equina syndrome; CS, corticosteroid; IVIG, intravenous immunoglobulin; PP, plasmapheresis; AAb, antiganglioside antibodies; NA, not applicable.

## 4. Conclusions

A few cases have been reported to develop GBS following elective spinal surgery. However, the occurrence of postoperative GBS is hard to detect in patients presenting together with cervical spinal cord injury. The time of symptom onset and the presence of lower motor neuron signs are crucial hints. Receiving only supportive care for GBS, this case failed to restore any walking ability even though his original AIS grading was D. To summarize, our case highlights the importance of not only early diagnosis of GBS, but also the importance of appropriate treatment being applied in a timely manner.

## Figures and Tables

**Figure 1 medicina-58-01063-f001:**
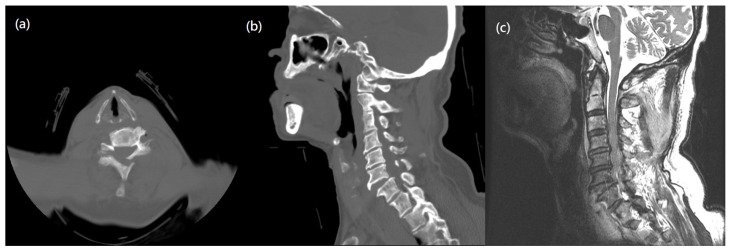
Pre-operative images. (**a**) Axial view of C spine CT showed a fracture in posterior element of C5-C6. (**b**) Sagittal view of C spine CT demonstrated traumatic spondylolisthesis of C6 on C7. (**c**) MRI T2-weighted image of C spine showed anterior subluxation of C6-7 with posterior HIVD, compressing the spinal cord, and soft tissue swelling in dorsal neck.

**Figure 2 medicina-58-01063-f002:**
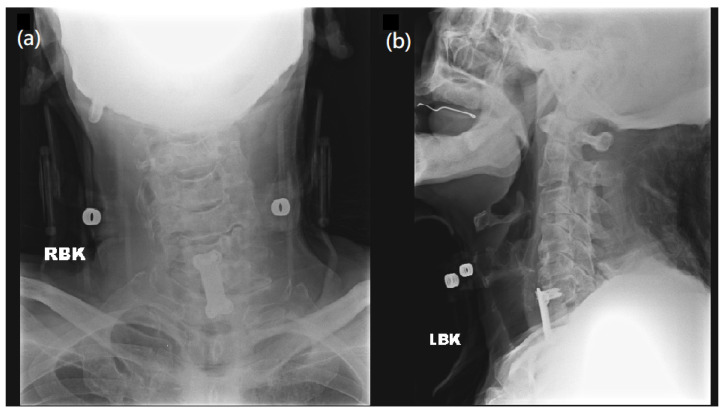
Postoperative radiography over C spine. (**a**) AP view and (**b**) Lateral view of C spine radiography demonstrates disc space narrowing over C4-5, C5-6, C6-7, and degenerative change plus anterior osteophyte formation, with cervical spine locking plates and screws over C6-T1, with no sign of instrumentation backout.

**Figure 3 medicina-58-01063-f003:**
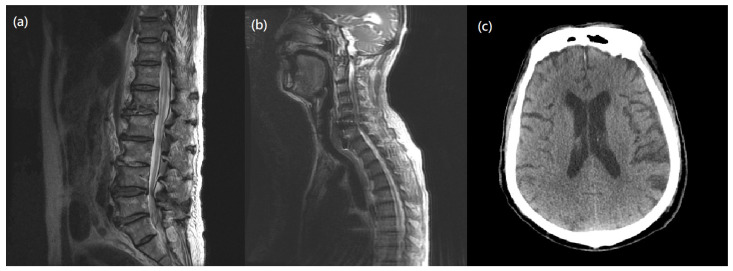
Postoperative Imaging. (**a**) MRI T2-weighted image of L spine showed L1 compression fracture, degenerative scoliosis without significant spinal stenosis or nerve root compression. (**b**) MRI T2-weighted image of cervical spine showed no new onset of intraspinal pathology. (**c**) Brain CT revealed no intracerebral hemorrhage, and no midline shift or space-occupying lesion.

**Figure 4 medicina-58-01063-f004:**
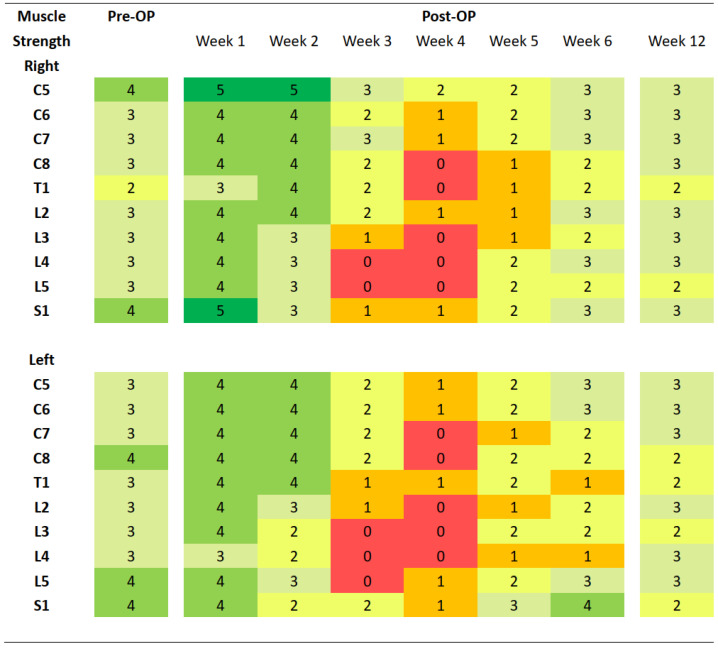
Chart recording muscle strength grading of the patient. Manual muscle strength is graded from 0 to 5 according to the Oxford Scale. Pre-OP, before operation; Post-OP, after operation.

**Table 1 medicina-58-01063-t001:** Summary of nerve conduction studies.

Nerve (Recording Site)	Stimulation Site	Amplitude Motor (mV) Sensory (μV)		Latency (ms)		Velocity (m/s)		F-Wave Latency (ms)	
		L	R	L	R	L	R	L	R
*Motor Conduction Study*									
Median (APB)	Wrist	2.8	3.0	4.0	4.2			NR	NR
	Elbow	2.5	2.7	8.7	9.1	50.1	49.2		
Ulnar (ADM)	Wrist	3.8	5.2	3.3	3.0			32.7	31.5
	Below Elbow	3.5	4.5	7.5	6.7	44.9	50.4		
	Above Elbow	3.3	4.4	9.6	8.7	48.5	51.1		
Tibial (AHB)	Ankle	3.6	1.5	3.7	4.0			53.1	54.5
	Knee	2.3	1.0	13.4	14.2	41.7	39.7		
Peroneal (EDB)	Ankle	0.3	0.1	4.0	4.4			NR	NR
	Below fibula	0.4	0.1	11.9	13.0	42.4	38.9		
*Sensory Conduction Study*									
Median (Index finger)	Wrist	9.5	12.3	2.7	2.9	52.1	48.5		
Ulnar (Little finger)	Wrist	21.1	20.1	2.5	2.4	55.1	57.4		
Radial (Snuffbox)	Forearm	12,1	13.7	2.3	2.2	46.5	47.2		
Sural (Posterior ankle)	Calf	4.7	5.6	3.5	3.1	41.2	46.7		
*H reflex*								H Latency (ms)	
Tibial (Soleus)	Knee							NR	NR

APB, abductor pollicis brevis; ADM, abductor digiti minimi; AHB, abductor hallucis brevis; EDB, extensor digitorum brevis; L, left; NR, no response; R, right.

**Table 2 medicina-58-01063-t002:** Summary of electromyography.

Muscle	Spontaneous			MUAP			Recruitment
	Fib	PSW	Fasc	Amplitude	Duration	Poly	
Right FDI	none	none	none	NL	NL	Few	Reduced
Right Biceps	none	none	none	NL	NL	NL	Reduced
Left FCR	none	none	none	NL	NL	NL	Reduced
Left APB	none	none	none	NL	NL	Few	Reduced
Left FDI	none	none	none	NL	NL	Few	Reduced
Right TA	none	none	none	No volitional activity			
Right GN-med	none	none	none	No volitional activity			
Left TA	none	none	none	No volitional activity			
Left GN-med	none	none	none	NL	NL	Few	Reduced

FDI, first dorsal interosseus; FCR, flexor carpi radialis; ABP, abductor pollicis brevis; EHL, extensor hallucis longus; GN-med, gastrocnemius (medial head); TA, tibialis anterior, FIB, fibrillation potentials; PSW, positive sharp waves; Fasc, fasciculations; Poly, polyphasia; NL, normal; MUAP, motor unit action potentials.

## Data Availability

Not applicable.

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
