# Peer review of "Guillain-Barré Syndrome with Respiratory Failure following Spine Surgery for Incomplete Cervical Cord Injury: A Case Report and Literature Review"

_medicina, 2022, doi:10.3390/medicina58081063_

Round 1
Reviewer 1 Report
No introduction was given.
Case Report -
There were no data about preceding infection, but what about diagnostics? Was the infection excluded (virus serology, Campylobacter jejuni)? How about antigangioside antibodies?
Electromyoneurography showed chronic axonal sensorimotor polyneuropathy. However, you concluded that it was acute polyneuropathy (GBS). This need further clarification, including table with NCS parameters.
Hyperoteinorachia can be present in other conditions, including compressive myelopathy. This also must be discussed.
Reviewer 2 Report
1. It is advised to the authors to read “instruction for authors”. The introduction should be revised because is not related to the main text.
2. A table with all the laboratory texts with normal ranges should be provided as supplementary material.
3. Could the authors provide the figures of the electrodiagnostic studies?
4. Description of brain MRI in the post-operative period should be done. Spinal MRI (Figure 3) should describe the weight (T1 or T2?).
5. Was the patient in use of any medication? Family history should be described.
6. EMG findings that are advised to be described: distal latency? amplitude? F waves?
7. A table with other cases with the literature is advised. Some variables: reference, age, sex, presentation, lesion, management, and outcomes.
8. Some opinions of authors need to: could this be a severe case of chronic inflammatory demyelinating polyradiculoneuropathy secondary to the trauma?
9. ‘‘Ethical review and approval were waived for this study due to the manuscript does not disclose the patient’s personal information.’’ This is not the right statement and should be removed from the manuscript.
Round 2
Reviewer 1 Report
Row 26 and 111 - acute on chronic. Does this mean subacute or maybe chronic changes following acute injury?
Reviewer 2 Report
Satisfactory.
Author Response
Dear reviewer,
Thank you very much for your valuable instructions, giving us a great opportunity to re-examinate our work. By addressing all your comments point-by point, we have improved our manuscript to a much better version.
Thanks again for your guidance and approval.
Sincerely,
Corresponding author:
Chun-Sheng, Hsu MD, PhD
Department of Physical Medicine and Rehabilitation, Taichung Veterans General Hospital, 1650, Section 4, Taiwan Boulevard, Xitun Dist., Taichung City 407, Taiwan.
Tel.: +886-423592525 ext. 3534
E-mail: chincent@vghtc.gov.tw